# Genome-Wide Association Analysis Identifies Candidate Loci for Callus Induction in Rice (*Oryza sativa* L.)

**DOI:** 10.3390/plants13152112

**Published:** 2024-07-30

**Authors:** Wintai Kamolsukyeunyong, Yeetoh Dabbhadatta, Aornpilin Jaiprasert, Burin Thunnom, Wasin Poncheewin, Samart Wanchana, Vinitchan Ruanjaichon, Theerayut Toojinda, Parichart Burns

**Affiliations:** National Center for Genetic Engineering and Biotechnology (BIOTEC), National Science and Technology Development Agency (NSTDA), Pathum Thani 12120, Thailand; yeetoh@biotec.or.th (Y.D.); aornpilin_nam@hotmail.com (A.J.); burin.thu@biotec.or.th (B.T.); wasin.pon@biotec.or.th (W.P.); samart.wan@biotec.or.th (S.W.); vinitchan.rua@biotec.or.th (V.R.); theerayuttoojinda638@gmail.com (T.T.)

**Keywords:** rice, GWAS, QTLs, callus induction, gene

## Abstract

Callus induction (CI) is a critical trait for transforming desirable genes in plants. A genome-wide association study (GWAS) analysis was conducted on the rice germplasms of 110 *Indica* rice accessions, in which three tissue culture media, B5, MS, and N6, were used for the CI of those rice panels’ mature seeds. Seven quantitative trait loci (QTLs) on rice chromosomes 2, 6, 7, and 11 affected the CI percentage in the three media. For the B5 medium, one QTL (qCI–B5–Chr6) was identified on rice chromosome 6; for the MS medium, two QTLs were identified on rice chromosomes 2 and 6 (qCI–MS–Chr2 and qCI–MS–Chr6, respectively); for the N6 medium, four QTLs were identified on rice chromosomes 6, 7, and 11 (qCI–N6–Chr6.1 and qCI–N6–Chr6.2, qCI–N6–Chr7, and qCI–N6–Chr11, respectively). Fifty-five genes were identified within the haplotype blocks corresponding to these QTLs, thirty-one of which showed haplotypes associated with different CI percentages in those media. qCI–B5–Chr6 was located in the same region as qCI–N6–Chr6.2, and the Caleosin-related family protein was also identified in this region. Analysis of the gene-based haplotype revealed the association of this gene with different CI percentages in both B5 and N6 media, suggesting that the gene may play a critical role in the CI mechanism. Moreover, several genes, including those that encode the *beta-tubulin protein*, *zinc finger protein*, *RNP–1 domain-containing protein*, and *lysophosphatidic acid acyltransferase*, were associated with different CI percentages in the N6 medium. The results of this study provide insights into the potential QTLs and candidate genes for callus induction in rice that contribute to our understanding of the physiological and biochemical processes involved in callus formation, which is an essential tool in the molecular breeding of rice.

## 1. Introduction

Callus induction is a process used in plant tissue culturing, in which plant cells, such as those from shoots, roots, and leaves, are stimulated to generate a mass of undifferentiated cells called calli [1,2]. This callus can be used for various purposes, such as genetic transformation, mutation breeding, and somatic embryogenesis in plant breeding programs. Callus induction is often used in rice breeding to introduce desired traits into rice plants through genetic modification or to regenerate whole plants from callus cultures. This technique allows breeders to rapidly propagate specific plants and create new rice varieties with desirable traits, such as disease resistance, drought tolerance, or increased yield [3]. The molecular mechanisms of callus formation include induction and repression: induction includes auxin, cytokinin, and wound induction, as well as the reacquisition of embryonic or meristematic fate; meanwhile, repression is used to inhibit cell wall biogenesis and certain epigenetic mechanisms [2]. Auxin and cytokinin are necessary for callus formation and can have synergistic, additive, or antagonistic functions, depending on the circumstances.

Genotypes and media composition play essential roles in callus induction; in particular, media composition, such as plant hormones, carbon sources, and organic components, greatly influence callus formation efficiency. 2,4–Dichlorophenoxyacetic acid (2,4–D) and 1–Naphthaleneacetic acid (NAA) are forms of auxin, while benzyl adenine (BA), zeatin, and kinetin are forms of cytokinin commonly used for callus induction and plant regeneration [2]. Callus induction and secondary metabolite production in grape cv. Syrah cultivars was better than that of cv. Chardonnay at 30 g/L sucrose in induction media, and was reduced in both cultivars at higher sucrose concentrations [3]. There were variations in callus growth, fresh weight, and regeneration among the three sorghum lines. Natural growth regulators, such as coconut water and synthetic growth regulators, also significantly impacted sorghum callus induction [4]. 

Rice is one of the most important food crops in the world. It is a staple food source for approximately half of the global population and provides caloric requirements and nutritional value in the form of dietary fiber, oryzanol, vitamins, and minerals [5]. Rice is in the Genus *Oryza* of Family Poaceae; only *Oryza glaberrima* and *O. sativa* are commonly cultivated and consumed. The latter is grown in Asia and can be divided into the subspecies *Indica* and *Japonica*. There have been studies on rice callus induction in mature *Japonica* [6,7,8] and *Indica* rice seeds [9,10], as well as comparisons among the concentrations of 2,4–D, NAA, BA, carbon sources, and basal media for rice callus induction and regeneration optimization in *Japonica* and *Indica* rice [11,12]. The effects of rice genotypes on callus induction were also investigated; for example, Singh et al. (2018) reported variations in the percentages of callus induction, cell growth, and morphology in four varieties of basmati rice (*Indica*) [13]. The synergistic effects of casein hydrolysate and 2,4–D on callus induction were reported among six *Indica* rice varieties [14]. The optimized concentrations of 2,4–D and BAP in induction media were found to be variable among four *Indica* rice cultivars [15]. Optimizing 2,4–D concentration and induction time enhanced callus proliferation, while the genotypes of the three Japonica rice varieties had no effect [16]. The callus induction rate has been reported to be higher in *Japonica* than in *Indica* rice varieties [17,18]. The challenge of callus induction in rice lies in accommodating the genetic background of the target rice variety and finding an optimized medium composition with which to obtain a high callus induction rate.

Identifying QTL/genes associated with callus induction could elevate the callus induction rate and reduce the time needed for applications in molecular selection and breeding new rice varieties. Wu et al. (2022) showed that *Pseudo-response Regulators* (*PRR*) located on rice chromosome 3 regulated rice callus formation [19]; this gene reportedly involved circadian clock components [20]. Taguchi-Shiobara et al. (2006) identified two QTLs, qIc4 and qIc9, associated with induced callus color, and the QTL qSc4, related to the colors of subcultured calli in BC_1_F_3_ lines derived from *Japonica* Koshihikari and *Indica* Kasalath rice [21]. Thirty QTLs associated with callus browning, a common phenomenon causing lower regeneration, poor multiplication, and death of the callus, have been mapped into chromosomes 1, 2, 3, 4, 8, 9, and 12 in Dongxiang wild rice [22]. One QTL, qcir9.1, associated with callus induction rate in the anther culture was identified on chromosome 9, with *Os090551600* encoding a high-mobility group (*HMG*) protein, most likely a candidate gene [23]. Guo et al. (2023) investigated gene expression changes during rice callus formation [24] and found that genes related to auxin signaling pathways, such as *auxin signaling F-box protein* (*AFR2*), *leaf cotyledon 1* (*LEC1*), and *WUSCHEL-related homeobox protein* (*WOX*), were highly expressed during scutellum-derived callus development. 

Genome-wide association studies (GWASs) using a massive number of single-nucleotide polymorphisms (SNPs) have become powerful tools for identifying genes that control important traits in plants [25]. GWASs allow researchers to scan the entire genome of a plant species to identify genetic variations associated with specific traits through high-resolution mapping that can pinpoint genes or genomic regions that influence critical agronomic traits, such as yield, disease resistance, and stress tolerance [26]. By studying diverse germplasm collections, novel alleles and genetic pathways that regulate critical agronomic traits can be uncovered [27]. Identifying specific genes or genomic regions associated with a trait of interest can elucidate the underlying genetic architecture and molecular mechanisms governing that trait. GWAS results can provide molecular markers linked to important traits, facilitating marker-assisted selection in plant breeding programs. Breeders can use these markers to efficiently introduce desirable traits into elite crop varieties through molecular breeding approaches [28]. GWAS analysis has revolutionized the field of plant genetics by providing a comprehensive and robust approach to identifying genes that control important traits in plants [29]. GWAS analysis of 510 rice accessions revealed 21 associated loci in rice callus induction QTLs, such as callus weight, color, and size [30]. Genes within these loci included *CRL1*, *OsBMM1*, *OsSET1*, and *OsIAA10*. The functions of *OsIAA10* include auxin perception and activation of downstream genes, such as *CRL1*, causing callus formation. GWAS analysis and functional annotation were used together to identify chromatin-enriched noncoding RNAs (cheRNAs) in rice calli that function in chromatin remodeling. The loss of function in the two cheRNAs resulted in damage to cell dedifferentiation and plant regeneration [31]. 

In this study, we performed GWAS analysis on the SNP dataset of 110 rice germplasm accessions, which were tested as having a 100% germination rate, to identify genomic regions associated with callus induction in the three following tissue culture media: B5, MS, and N6 [32,33,34]. Candidate genes and their possible functions in callus induction were proposed.

## 2. Results

### 2.1. Seed Germination and the Formation of Scutellum-Derived Calli in the Rice Germplasm Collection

In order to effectively evaluate the callus induction rate of rice accessions, it was necessary to eliminate factors that may affect callus induction, such as the germination rate of the seeds used. In this study, 289 rice accessions collected at the Rice Science Center, Kasetsart University, Thailand, were tested for their germination rates (Appendix A); only 110 rice accessions had a 100% germination rate and were therefore selected for the callus induction experiment, in which the scutellum-derived calli that emerged from the mature seeds were evaluated. Embryogenic calli normally occur 2–3 weeks after cultivation in the medium. The response to callus induction was evaluated at three different levels of callus formation (Figure 1), and the callus induction rate was calculated based on partial and complete embryogenic calli.

Typically, callus initiation from the scutellum of a mature seed depends on the levels of hormones and the components of the callus induction medium; moreover, the genetic dependence effect also affects the callus induction rate among cultivars. In order to evaluate the impacts of hormones and the components in the callus induction medium, we assessed the callus induction (CI) traits of the 110 selected rice accessions in the three following tissue culture media: B5, MS, and N6 (Appendix A). When all 110 rice varieties were grown in these three types of solid media for three weeks, all rice accessions successfully developed calli in the different media formulations at different rates. The CI percentages among the 110 accessions averaged 25.43%, 33.56%, and 28.42% for the B5, MS, and N6 media, respectively (Appendix A). A moderate correlation was found among the three media (Figure 2). Among the 110 rice accessions, two rice varieties, including Niaw Ubon 1 and Chum Phae 60, showed high CI (≥60%CI) in all three media (Niaw Ubon 1 = 77.8, 86.7, and 60.0; Chum Phae = 62.2, 73.3, and 60.0 for the B5, MS, and N6 media, respectively). One rice variety, Kiaw, showed a high CI in B5 (64.4) and MS (66.7) media, and one variety, Luang Thong, showed a high CI in MS (73.3) and N6 (64.4) media. In addition, JHN–5 showed the highest CI in the N6 medium (91.1%). In the N6 solid medium, 38 rice accessions exhibited callus induction percentages greater than 50%. Jao Hom Nin (JHN) showed the highest callus induction, at 97.78% ± 3.85, followed by Phitsanulok 60–1 (84.44% ± 3.85), HomMali801 (77.78% ± 10.18), and Homnaypol (76.67% ± 4.71) (Appendix A). In the B5 solid medium, 27 rice accessions demonstrated a high percentage of callus formation, with induction rates ranging between 50% and 78%, though these differences were not statistically significant. For the rice accessions grown in MS solid medium, 48 rice accessions exhibited callus percentages ranging from 50% to 89% without statistically significant differences.

### 2.2. Genotype Data and Analysis of Population Structure, Relative Kinship, and Linkage Disequilibrium (LD)

In this study, we used 2,385,475 single-nucleotide polymorphisms (SNPs) as genotype data, averaging 198,790 SNPs per chromosome. The SNP density on each chromosome is shown in Figure 3A. All of the SNPs had a minor allele frequency (MAF) greater than 0.05. To investigate the population structure and cryptic relationships among the 110 rice accessions, a principal component analysis (PCA) and kinship matrix were employed, revealing some degree of population structure within the rice diversity panel (Figure 3B, Appendix A). The majority of the rice accessions in the panel, which included landraces and improved varieties from Thailand, could be classified into a single large group (Figure 3C).

Linkage disequilibrium (LD) decay analyses were conducted in order to evaluate chromosomal signatures of recombination patterns (Figure 4). The mean LD decay values within intrachromosomal distance bins dropped below a threshold mean r^2^ value of 0.2 between 70 and 259 kb across all 12 chromosomes, with an average of 163.75 kb. The LD decay signatures were similar for all chromosomes except for chromosome 11, which exhibited a markedly lower threshold crossover point of 70 kb.

### 2.3. Identification of QTLs Associated with Callus Induction Using GWAS

We performed a genome-wide association study (GWAS) on 110 rice accessions to identify genomic regions associated with callus induction (CI) in rice. GWAS analysis, featuring a multi-locus mixed-model analysis (MLMM), identified seven regions associated with CI traits (Figure 5) located on chromosomes 2, 6, 7, and 11. Regions containing adjacent associated SNPs within an LD block (r^2^ > 0.2) were combined, resulting in seven QTLs: qCI–B5–Chr6, qCI–MS–Chr2, qCI–MS–Chr6, qCI–N6–Chr6.1, qCI–N6–Chr6.2, qCI–N6–Chr7, and qCI–N6–Chr11 (Table 1).

### 2.4. Haplotype Block Analysis and Candidate Gene Identification

Candidate genes within each QTL were identified using adjusted LD blocks, inferred significant variants, and gene-based haplotype analyses. The number of candidate genes differentiating between phenotypes varied from 0 to 16 per QTL (Table 2). The highest number of candidate genes was identified in the B5 medium, whereas none were found in the MS medium. Notably, the gene *Os06g0254600*, which encodes a *Caleosin-related family protein*, was identified in the B5 and N6 media, corresponding to qCI–B5–Chr6 and qCI–N6–Chr6.2, respectively.

For the qCI–B5–Chr6 region, five haplotype blocks were defined within the 475 kb flanking region, spanning from 7.915 to 8.390 Mb, based on the analysis of 1390 SNPs. Within this region, 28 genes were annotated (Appendix A), and 16 genes were explicitly identified within the candidate haploblock between 7.934 and 8.198 Mb (Figure 6A, Table 2).

Using gene-based haplotype analysis, all sixteen genes were found to contain haplotypes associated with different callus induction (CI) levels. Notably, the gene *Os06g0256600* (Figure 6B) exhibited the most significant association, with two haplotypes, Hap I (n = 49) and Hap V (n = 4), associated with lower CI percentages (averaging 21.18% and 22.78%, respectively), while Hap III (n = 11) and Hap IV (n = 8) were associated with higher CI percentages (averaging 37.07% and 36.39%, respectively). Thus, this gene was annotated as a conserved hypothetical protein.

Genes *Os06g0255200* and *Os06g0254200* each contained three haplotypes, showing a similar pattern in their association with CI (Figure 6C,D). Hap I in both genes was consistently associated with a lower CI percentage (averaging 22.57% and 22.33%, respectively), while Hap II was associated with a higher %CI (averaging 35.51% for both genes). *Os06g0255200* is annotated as a gene encoding a *conserved hypothetical protein* involved in chromatin remodeling ATPase function and potentially influencing embryo development. Meanwhile, *Os06g0254200* was annotated as a gene similar to the *potassium channel protein NKT5*. Interestingly, two genes annotated to encode *Caleosin-related family proteins*, *Os06g0254300* and *Os06g0254600*, known for their role in lipid degradation during seed germination, were identified to have haplotypes associated with CI.

For the MS medium, two QTLs were identified: qCI–MS–Chr2 and qCI–MS–Chr6. The first QTL, qCI–MS–Chr2, spans a 287 kb region from 8.817 to 9.104 Mb on chromosome 2 (Figure 7A). Within this region, thirty genes were annotated (Appendix A), and four genes were located in the candidate haploblock between 8.906 and 8.968 Mb. However, none of these four genes showed haplotypes significantly associated with different levels of CI, although *Os02g0258000*, the gene annotated to encode a *hypothetical conserved protein*, showed the highest non-significant association with a different CI of 7.30% between Hap I (average 33.70%CI; n = 87) and Hap II (average 39.04%CI; n = 7). The second QTL, qCI–MS–Chr6, is located in the 477 kb region from 16.575 to 17.052 Mb on chromosome 6 (Figure 7B), a region that contains twenty-six annotated genes (Appendix A). Within the 406 kb spanning from 16.575 to 16.981 Mb, thirteen annotated genes were identified. Unfortunately, similar to those found in qCI–MS–Chr2, none of these genes exhibited haplotypes significantly associated with varying CI levels; however, four genes located in this region showed the highest non-significant association with a different CI of 16.35% between Hap III (average 48.15%CI; n = 3) and Hap II (average 31.80%CI; n = 21) in every gene. These genes were annotated as *OsMLO6* (*Os06g0486300*), encoding a *serine/threonine protein kinase domain-containing protein* (*Os06g0486400*), a *hypothetical protein* (*Os06g0487300*), and a *protein of unknown function DUF1604 domain-containing protein* (*Os06g0489200*).

For the N6 medium, four QTLs were identified, including qCI–N6–Chr6.1 and qCI–N6–Chr6.2 on chromosome 6, qCI–N6–Chr7 on chromosome 7, and qCI–N6–Chr11 on chromosome 11. The first of these QTLs, qCI–N6–Chr6.1, spans the 475 kb region between 3.266 and 3.741 Mb on chromosome 6 (Figure 8A), encompassing fifty-three annotated genes (Appendix A), with four genes located in the candidate haploblock between 3.483 and 3.555 Mb, including *Os06g0169600*, *Os06g0169800*, *Os06g0169900*, and *Os06g0170500*; only *Os06g0169600*, *Os06g0169800*, and *Os06g0170500* contained haplotypes associated with different CI percentages (Figure 8B,C). Hap I (n = 85, 76, and 86, respectively) was associated with lower CI percentages (averaging 25.96%, 25.56%, and 26.06%, respectively), while Hap II (n = 17 for all three genes) was associated with higher %CI (40.92%, on average, for all three genes). These genes were annotated as encoding the *protein similar to beta-tubulin* (*Os06g0169600*), the *hypothetical protein* (*Os06g0169800*), and similar to *RNA-binding protein* (*Os06g0170500*), respectively.

The second QTL, qCI–N6–Chr6.2, is also located on chromosome 6, comprising a 476 kb region between 7.766 and 8.242 Mb (Figure 9A), which contained thirty-one annotated genes (Appendix A), but only one candidate gene, *Os06g0254600*, was identified within the candidate haploblock between 7.995 and 8.014 Mb. Gene-based haplotype analysis revealed that *Os06g0254600*, which encodes a *Caleosin-related family protein*, had haplotypes associated with different CI levels. Hap I (n = 72) was associated with a lower %CI, averaging 24.57%, while Hap II (n = 20) and Hap III (n = 3) were associated with higher CI percentages, averaging 41.39% and 41.48%, respectively (Figure 9B).

The third QTL, qCI–N6–Chr7, was identified within a 248 kb region between 8.620 and 8.868 Mb (Figure 10A), containing seven annotated genes (Appendix A). Five of these genes, located within a 124 kb candidate haploblock from 8.734 to 8.858 Mb, included *Os07g0255900* and *Os07g0256700* (annotated as the genes with *the domain of unknown function DUF231*), *Os07g0256300* and *Os07g0256866* (the genes encoding *hypothetical proteins*), and *Os07g0256200* (the gene with the *RNA recognition motif*, *RNP–1 domain-containing protein*). Gene-based haplotype analysis revealed that only two genes, *Os07g0256200* and *Os07g0256866*, had haplotypes associated with different CI levels in the N6 medium (Figure 10B,C). For *Os07g0256200*, among the four haplotypes, Hap IV was associated with a higher %CI (average 47.78%), while Hap III was associated with a lower %CI (average 20.87%). The other two haplotypes, Hap I and Hap II, were associated with medium CI levels (averages of 27.81% and 28.38%, respectively). For *Os07g0256866*, three haplotypes were identified, with Hap III associated with higher %CI (average 50.99%) and Hap I and Hap II with lower %CI (average 25.76 and 27.42%, respectively).

The last QTL of the N6 medium, qCI–N6–Chr11, was situated within a 136 kb region spanning from 25.108 to 25.244 Mb (Figure 11A); within this region, seventeen genes were annotated (Appendix A), with thirteen genes forming the candidate haploblock between 25.148 and 25.223 Mb. Gene-based haplotype analysis identified ten genes whose haplotypes were associated with different CI levels in the N6 medium, including *Os11g0637050*, *Os11g0637900*, and *Os11g0638200* (annotated as the genes encoding a *hypothetical protein*); *Os11g0637000*, *Os11g0637100*, and *Os11g0637200* (encoding *monosaccharide transporter (PLT subfamily)*, *PLT protein 10*); *Os11g0637600* (encoding *protein similar to potyvirus VPg interacting protein*); *Os11g0637700* (encoding *RNA-binding protein*); and *Os11g0637800* (*Lysophosphatidic acid acyltransferase 2*). Among these, *Os11g0637700* exhibited the most significant difference in CI associated with its haplotypes (Figure 11B), in that Hap III (n = 7) was associated with a higher %CI (average 50.48%), while Hap I (n = 51) was associated with a lower %CI (average 21.85%). Similarly, *Os11g0637800* also showed notable variation in CI between haplotypes (Figure 11C); Hap I (n = 95) was associated with a lower %CI (average 26.89%), whereas Hap II (n = 7) was associated with a higher %CI (average 50.48%).

## 3. Discussion

In this study, we found that the three researched media, B5, MS, and N6, combined with 2,4–D hormone at a concentration of 2 mg/L, had varying levels of efficiency for inducing callus formation in rice, influenced by several other factors. Josefina and Kazumi (2010) observed differences in callus formation characteristics among *Indica*, *Japonica*, and *Javanica* rice groups [35]; in addition, 2,4–D has been reported as the most popular and effective growth regulator among auxins for inducing monocot tissues to form calli [36]. Callus formation in Jasmine rice and Pathum Thani 1 could be induced in media supplemented with 2,4–D at a 2 mg/L concentration [37]. As for comparing the three basal media (N6, B5, and MS), which are different in their levels of macronutrients, especially nitrogen [38], the percentage of callus induction can be evaluated from the number of callus formations among plants; although the characteristics of the calli formed in each basal medium were not different, the results clearly show the diverse callus formation types among the three basal media a variation that might explain the genetic dependence effect on responsiveness to the levels of macronutrients in the basal medium [39].

The genetic regulation of callus induction in rice involves multiple genes and pathways [40]. Several essential genes have been identified that are crucial in controlling callus formation and regeneration in rice, including *OsLEC1* (*LEAFY COTYLEDON1*), which is involved in maintaining embryonic characteristics in callus cells, essential for somatic embryogenesis in rice [24]; *OsWOX11* (*WUSCHEL-RELATED HOMEOBOX 11*), a crucial regulatory gene of stem cell maintenance, vital for callus induction and subsequent shoot regeneration in rice [41]; *OsPLT* (*PLETHORA*) genes, members of the *OsPLT* family involved in maintaining stem cell populations in root and shoot meristems, crucial for callus induction and regeneration processes [42]; and *OsARF* (*AUXIN RESPONSE FACTOR*) and *OsAux/IAA* (*AUXIN/INDOLE–3–ACETIC ACID*), genes involved in auxin signaling pathways that centrally regulate cell division and differentiation during callus formation in rice [43].

Genome-wide association studies (GWASs) have identified several candidate genes associated with callus induction in rice [30]. *DRO1* (*DEEP ROOT 1*) has been associated with enhanced root growth and development, potentially influencing callus induction efficiency by facilitating better nutrient and hormone uptake [44]. *OsRRMh* (*Rice RECEPTOR-LIKE PROTEIN KINASE HOMOLOG*) influences rice’s callus induction and regeneration processes by regulating stress responses and hormone signaling pathways [45,46]. *OsCYP94C2* (*Rice CYTOCHROME P450 MONOOXYGENASE*) plays a role in plant hormone biosynthesis and metabolism, particularly affecting auxins and cytokinins crucial for callus induction and regeneration [47]. *OsGH3.8* (*GRETCHEN HAGEN3.8*) contributes to auxin homeostasis and signaling, essential for regulating cell division and differentiation during callus formation in rice [48].

B5 medium is a commonly used plant tissue culture medium that supports the growth and development of various plant species. The B5 medium contains essential nutrients, vitamins, and growth regulators required for plant cell growth and differentiation. In this study, two candidate genes in qCI–B5–Chr6, *Os06g0254300* and *Os06g0254600*, were related to *Caleosin*; *Os06g0255200* and *Os06g0255700* were the genes *encoding chromatin-remodeling proteins*. Chromatin remodelers, such as CHD chromatin remodeler PKL, prevent induced callus formation; the *Arabidopsis* mutants with defective chromatin remodeling ATPase are hypersensitive to cytokinin and display enhanced callus greening [49,50]. *Os06g0256000* was identified as the gene that encodes *Condensin complex subunit III*. *Condensin–2 complex subunit H2* expression significantly increases during embryogenic cotton transdifferentiation [51] and can function similarly in rice calli.

For CI in MS medium, we identified that two genomic regions on rice chromosomes 2 and 6 were associated with different %CI values in rice accessions; however, no annotated genes containing haplotypes associated with %CI in rice accessions were identified in these two genomic regions. Our failure to find candidate genes may be because the population size used in this study was too small to detect a statistically significant association between the haplotype and the trait [52], or the trait may be complex and influenced by multiple genes and environmental factors [53]. Moreover, CI results in MS medium may be influenced by epigenetic modifications rather than a single gene [54]. Nevertheless, five potential genes that showed non-significant associations with different CI levels were proposed in these two genomic regions, and future studies should focus on these five candidate genes to confirm their roles in rice callus induction in MS medium.

The N6 medium is a specialized plant tissue culture medium particularly suited for the culture of cereal grains like rice [34]. Rice callus induction in the N6 medium is a valuable tool for rice improvement, enabling rapid propagation and genetic modification. In this study, four genomic regions on rice chromosomes 6, 7, and 11 were associated with different %CI of rice accessions in the N6 medium. For qCI–N6–Chr6.1, *Os06g0169600*, annotated as the gene encoding *beta-tubulin-containing GTPase domain*, was found to have a haplotype associated with different %CI. *Beta-tubulin* is a protein essential for forming microtubules for cell division and growth [55] and is involved in callus induction in soybeans [56]. Studies have shown that the expression of *beta-tubulin* genes increases in plant cells during callus induction, suggesting that *beta-tubulin* plays a role in cell division and growth. *Os06g0170500*, annotated as *OsC3H40*, encodes *zinc finger proteins* containing a motif with three cysteines and one histidine residue [57]. Expression analysis indicated that the genes in this family are regulated by abiotic or biotic stresses, suggesting that they could have influential roles in stress tolerance [58]; moreover, the gene was expressed in roots, flowers, and seeds. The role of *OsC3H40* in callus induction has not been previously reported; therefore, the functional impact of this gene on callus induction in rice remains to be confirmed.

For qCI–N6–Chr7, the following two candidate genes with haplotypes associated with different CI levels were identified: *Os07g0256200* and *Os07g0256866*. While *Os07g256866* was annotated as the gene encoding a *hypothetical protein* whose function was not reported, *Os07g0256200* was recently cloned as *OsTIF1* (*tilling number 1 (TN1) interaction factor 1*) [59], which positively regulates D14 expression and modulates axillary bud outgrowth, ultimately affecting tiller development; however, the role of the *OsTIF1* gene in callus induction still remains to be elucidated.

For qCI–N6–Chr11, ten candidate genes with haplotypes associated with different CI levels were identified. *Os11g0637700*, whose haplotypes showed the highest CI difference among the ten candidate genes, was annotated as the *nucleotide-binding*, *alpha–beta plait domain-containing protein* found to play a crucial role in callus development in *Arabidopsis* [60]. Another interesting candidate gene was *Os11g0637800*, annotated as the *lysophosphatidic acid acyltransferase 2* (*OsLPAT2*) gene, which is essential for salt and drought stress tolerance in rice [61]. Phosphatidic acid (PA) is an important signaling molecule involved in various cellular processes, including plant development, stress responses, and callus induction. *OsLPAT2* may influence callus induction through several mechanisms, such as cell signaling pathways, influencing membrane stability, interacting with auxin, ensuring the proper distribution and activity of auxin within cells, enabling cells to cope with stress conditions, and maintaining optimal growth for callus induction.

qCI–B5–Chr6 was located in the same region as qCI–N6–Chr6.2; this region may be considered a promising QTL for universal callus induction. One annotated gene, *Os06g0254600*, has a haplotype associated with %CI in both media and encodes *Caleosin-related family protein*. *Caleosin proteins* have been shown to play crucial roles in mediating lipid droplet degradation via microautophagy during seed germination in *Arabidopsis* [62]; in rice, a *Caleosin family* gene, *OsClo5* (*Os04g0511200*), has been identified recently [63] and was found to be involved in cold response via the jasmonic acid signaling pathway. The role of this gene in the callus induction process in rice tissue cultures remains to be elucidated.

## 4. Materials and Methods

### 4.1. Association Mapping Panel Germination Test and Callus Induction Evaluation

A collection of 289 diverse rice accessions from the germplasm of rice varieties in Thailand and other countries was used in this study. In order to evaluate the germination rate, ten mature seeds of each accession were randomly selected and cultured in solid MS medium plus 3% sucrose (pH = 5.8) under 16 h of light and 8 h of darkness for three weeks. Accessions with 100% germination rates were selected and used in further experiments. The callus induction experiment was conducted as follows: Seeds of the selected accessions were husked and then surface-sterilized for 5 min in sterile distilled water, a process that was repeated 3–4 times, followed by a 5-min rinse in 70% ethanol. The seeds were then shaken in 20% sodium hypochlorite for 20 min, followed by an additional 10 min, and then rinsed three times in sterile distilled water, dried on clean filter paper in a sterile Petri dish, and cultured in N6, B5, or MS medium. The media were supplemented with 2 mg/L 2,4–D, 2.87 g/L L-proline, 30 g/L sucrose, and 3.0 g/L phytagel, with the pH adjusted to 5.8 before autoclaving. Seeds were maintained in the dark at 25 ± 2 °C for three weeks. Each treatment included 15 seeds and three replicates. After three weeks, the callus induction frequency was recorded by counting the number of seeds that formed a partial and compact scutellum-derived callus.

### 4.2. Genome-Wide Association Analysis

GWAS was carried out on 2,385,475 SNP markers using GAPIT (Genome Association and Prediction Integrated Tool) software version 3 [64]. The analysis employed various models, including GLM, MLM, CMLM, MLMM, and FarmCPU. SNP data were derived from a whole-genome re-sequencing project at the National Center for Genetic Engineering and Biotechnology (BIOTEC), Thailand. The SNPs used in the GWAS analysis were homozygous, with a call rate of 90% and minor allele frequency (MAF) greater than 5%. They were aligned using the Nipponbare IRGSP 1.0 rice reference genome. Principal component (PC) and kinship analyses were performed using TASSEL based on LD-pruned SNPs to generate PC and kinship matrices. The STRUCTURE algorithm [65] was employed with a model incorporating admixture and correlated allele frequencies. Three independent replicates were conducted for each genetic cluster (K) value, ranging from K = 1 to K = 8, with a burn-in of 10,000 steps and a run length of 10,000 Markov Chain Monte Carlo (MCMC) iterations. PCA plots and kinship heatmaps were generated using GAPIT [66]. The SNP density of each chromosome was visualized using the R package RIdeogram [67]. Significantly associated SNPs were identified using a threshold of −log10 (*p*-value) ≥ 5.0, based on a formula that considers marker-based heritability [68].

### 4.3. Haplotype Block Analysis and Candidate Gene Identification

The haplotype block analysis for all QTLs was conducted using LDBlockShow, applying an LD decay rate threshold of 0.2 for each chromosome [69]. The blocks were subsequently adjusted to align with the identified QTLs. Within these blocks, SNPs were filtered to retain only the significant variants, utilizing the Ensembl Variant Effect Predictor (VEP) and focusing on the following six specific variant types: frameshift variants, in-frame insertions/deletions, stop-gained mutations, missense variants, and splice region variants [70].

The SNPs for each identified gene were then employed to construct haplotypes for phenotype differentiation using the geneHapR package in R [71]. The annotations of genes located within each QTL region were identified using the database from the Rice Annotation Database (https://rapdb.dna.affrc.go.jp) accessed on 26 July 2024 [72]. Candidate genes were determined based on their biological relevance.

## 5. Conclusions

Callus induction is a pivotal trait for transforming desirable genes into plants; in this work, we conducted a genome-wide association study (GWAS) on 110 diverse *Indica* rice accessions using three medium types: B5, MS, and N6. Seven quantitative trait loci (QTLs) on chromosomes 2, 6, 7, and 11 significantly affected the callus induction rate across these media. The results of this study provide insights into potential QTLs and candidate genes for callus induction in rice; however, the proposed candidate genes require further investigation. Once the most likely candidate genes are identified, we will develop functional markers for the callus induction trait and test their efficiency levels in segregated populations; their expression during callus development will also be revealed. qCI–B5–Chr6 was located in the same region as qCI–N6–Chr6.2; thus, this region may be considered promising for callus induction in general. The candidate gene that haplotypes associated with different %CI in both media within this region is *Os06g0254600*, which encodes a *Caleosin-related family protein*; however, further research is needed to confirm the roles of *Os06g0254600* in callus induction and unravel the complex genetic networks that control callus induction and regeneration in rice.

## Figures and Tables

**Figure 1 plants-13-02112-f001:**
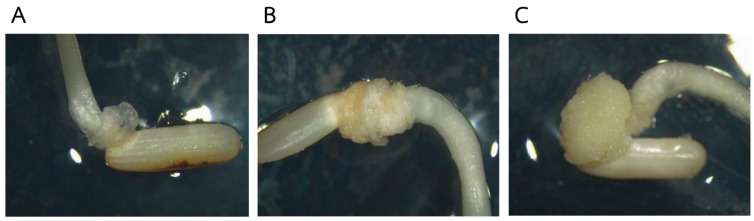
The different types of scutellum-derived callus formation: no callus induction (**A**), partial callus induction with incomplete embryogenic callus (**B**), and complete embryogenic callus induction (**C**).

**Figure 2 plants-13-02112-f002:**
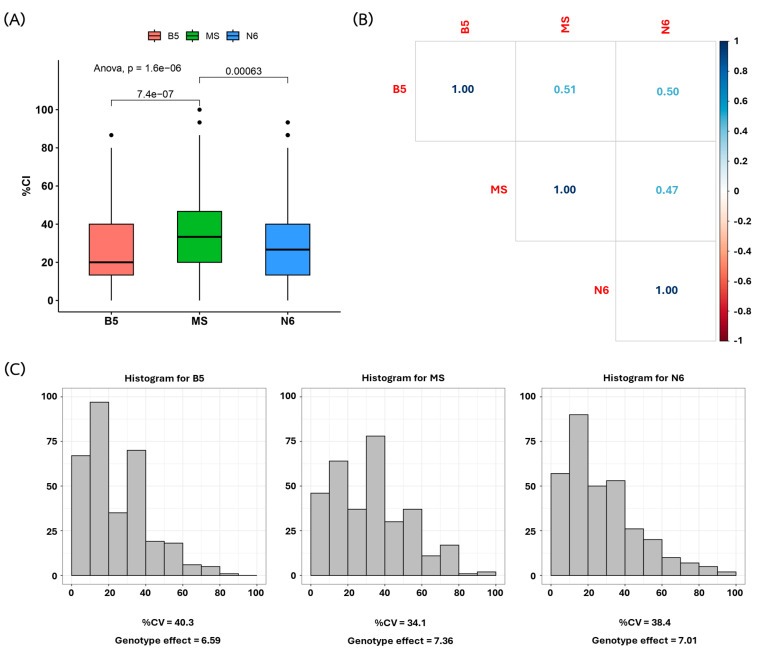
The callus induction (CI) percentages of the 110 rice accessions were evaluated across three tissue culture media: B5, MS, and N6. (**A**) The boxplot illustrates the percentage CI distribution analyzed using ANOVA and Tukey’s HSD test. (**B**) Correlation plots were generated to examine the relationships among the different media. (**C**) The histogram shows the distribution of the CI percentages in each medium.

**Figure 3 plants-13-02112-f003:**
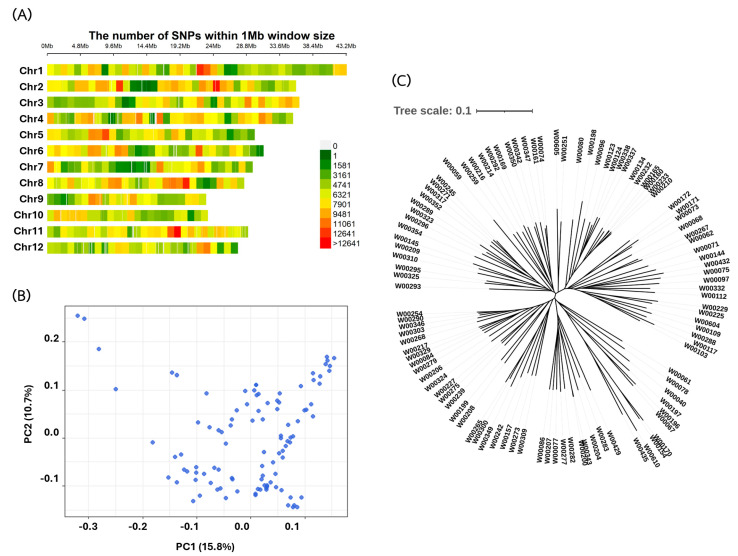
Genetic structure of the panel of 110 rice accessions: (**A**) SNP density on rice chromosomes, (**B**) principal component analysis (PCA) and kinship relatedness analysis of the 110 genotypes, and (**C**) population structure of the 110 rice accessions.

**Figure 4 plants-13-02112-f004:**
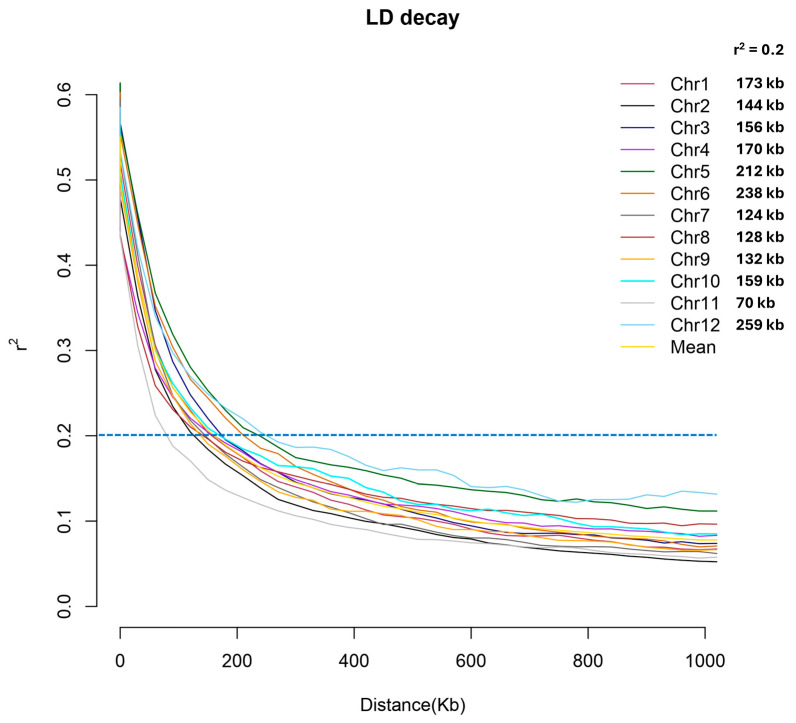
Linkage disequilibrium decay across 12 chromosomes in 110 *O. sativa* accessions. Mean LD decay ranges between 70 and 259 kb.

**Figure 5 plants-13-02112-f005:**
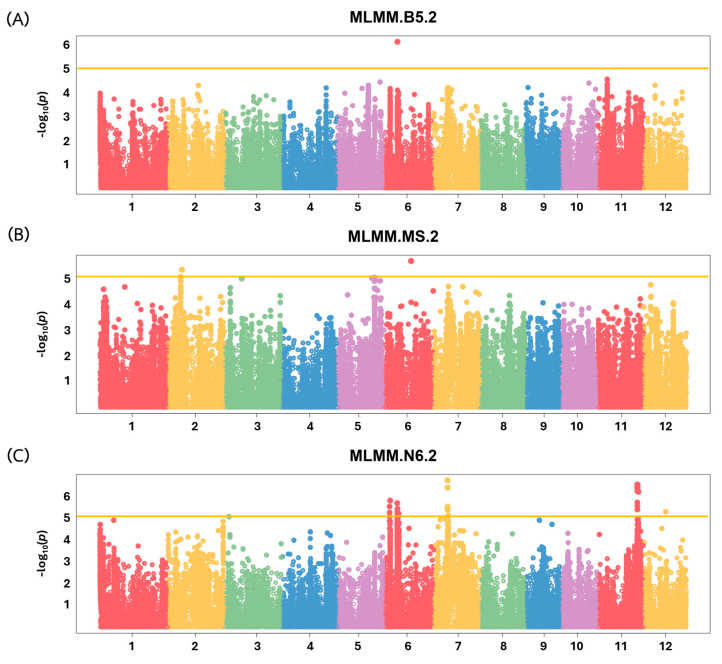
Manhattan plots resulting from genome-wide association study (GWAS) results for CI in three tissue culture media: (**A**) B5, (**B**) MS, and (**C**) N6. Yellow lines indicate the cut-off threshold at −log_10_ (p) of 5.0.

**Figure 6 plants-13-02112-f006:**
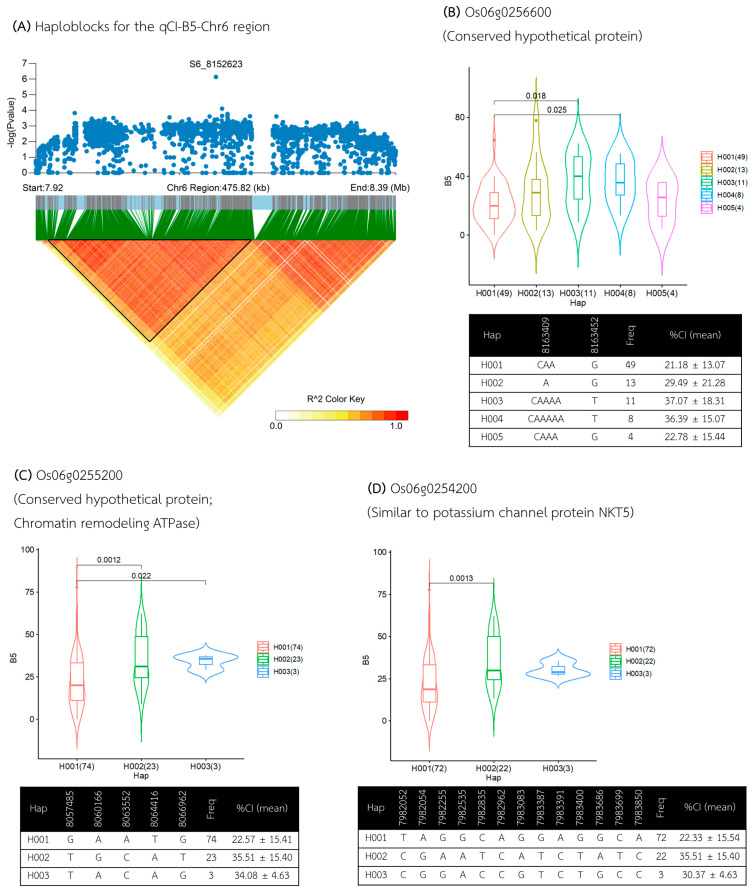
Analysis of associated regions and haplotype analysis within the qCI–B5–Chr6 region. (**A**) Manhattan plots and LD heatmap across the 475 kb region surrounding the significant SNP. The black triangle in the LD heatmap indicates the candidate haploblock. (**B**–**D**) Boxplots showing the distribution of %CI and haplotype analysis for the genes *Os06g0256600*, *Os06g0255200*, and *Os06g0254200*.

**Figure 7 plants-13-02112-f007:**
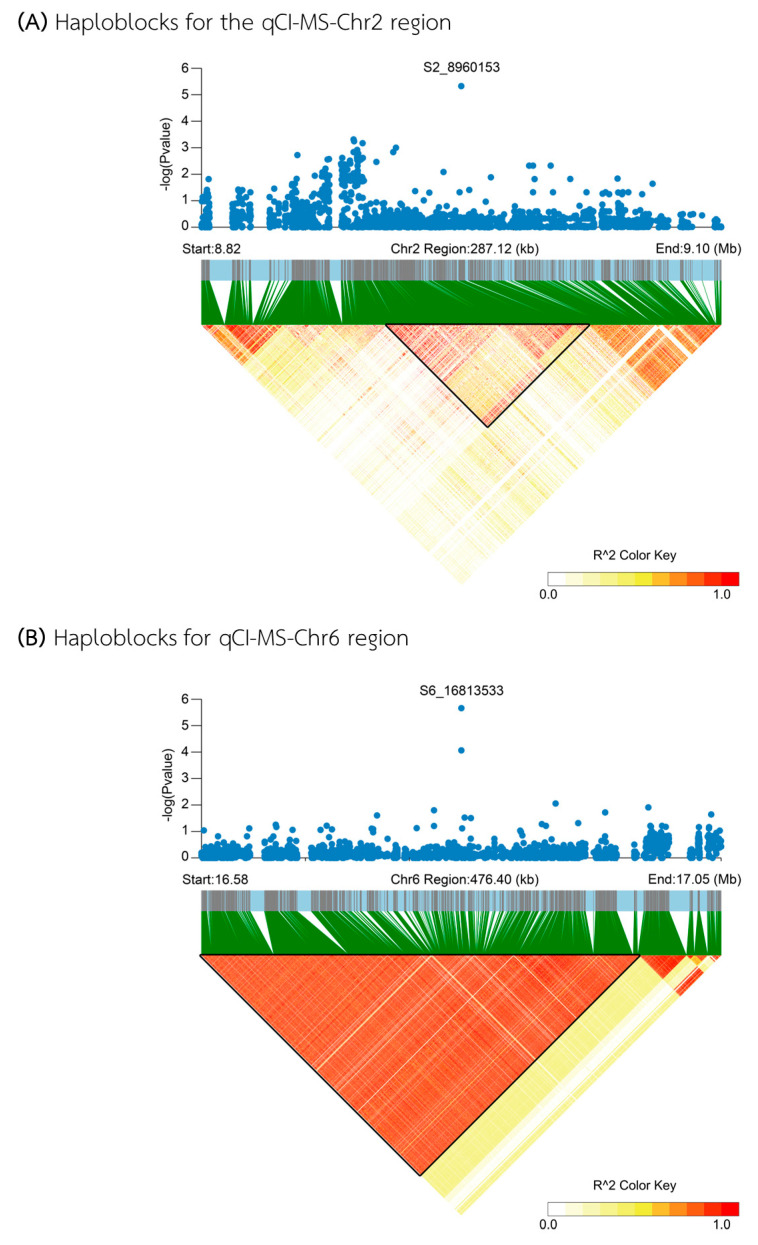
Analysis of associated regions and haplotypes within the qCI–MS–Chr2 (**A**) and qCI–MS–Chr6 (**B**) regions. The black triangles in the LD heatmaps indicate the candidate haploblock.

**Figure 8 plants-13-02112-f008:**
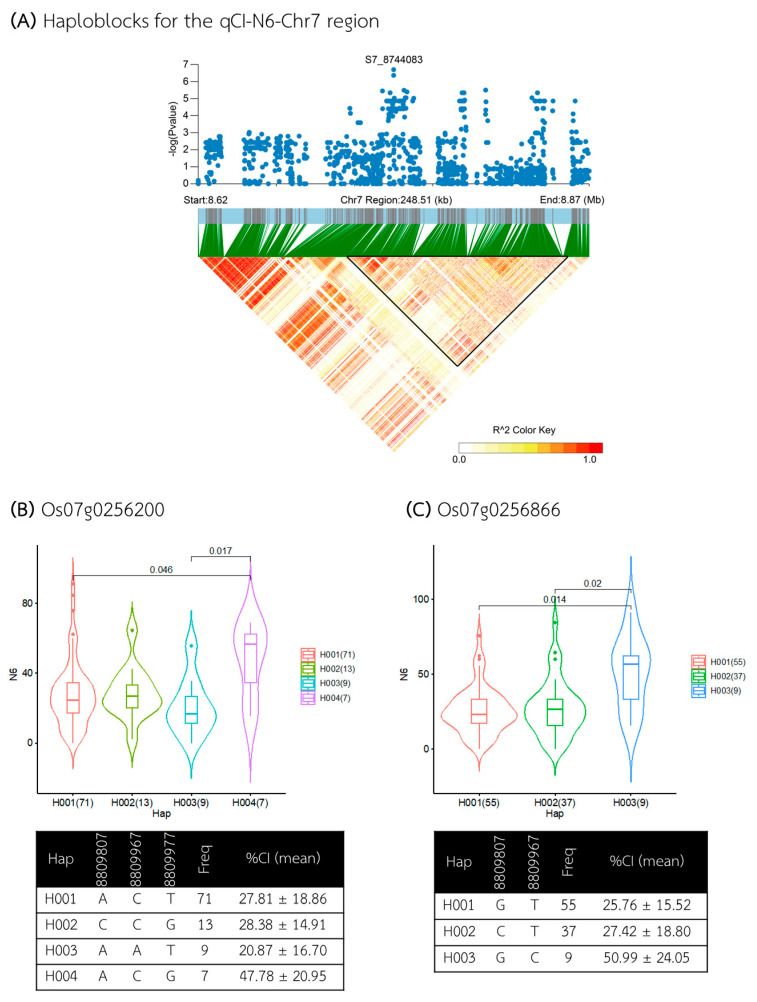
Analysis of associated regions and haplotypes within the qCI–N6–Chr6.1 region. (**A**) Manhattan plot and LD heatmap across the 475 kb region surrounding the significant SNP. The black triangle in the LD heatmap indicates the candidate haploblock. (**B**,**C**) Boxplots showing the distribution of %CI and haplotype analysis for the genes *Os06g0169600*, *Os06g0169800*, and *Os06g0170500*.

**Figure 9 plants-13-02112-f009:**
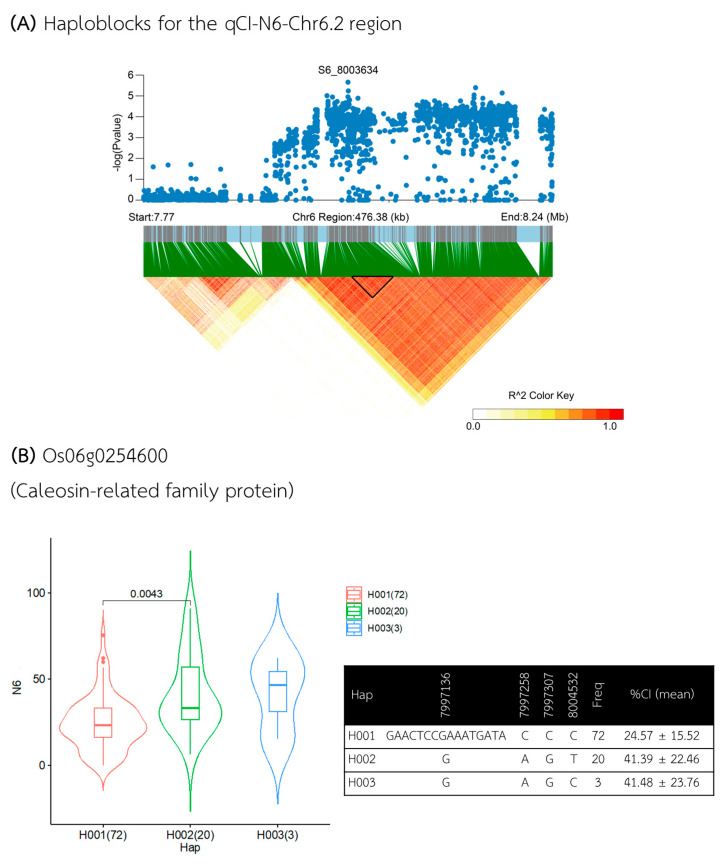
Analysis of associated regions and haplotype analysis within the qCI–N6–Chr6.2 region. (**A**) Manhattan plot and LD heatmap across the 476 kb region surrounding the significant SNP. The black triangle in the LD heatmap indicates the candidate haploblock. (**B**) Boxplot showing the distribution of %CI and haplotype analysis for the Os06g0254600 gene.

**Figure 10 plants-13-02112-f010:**
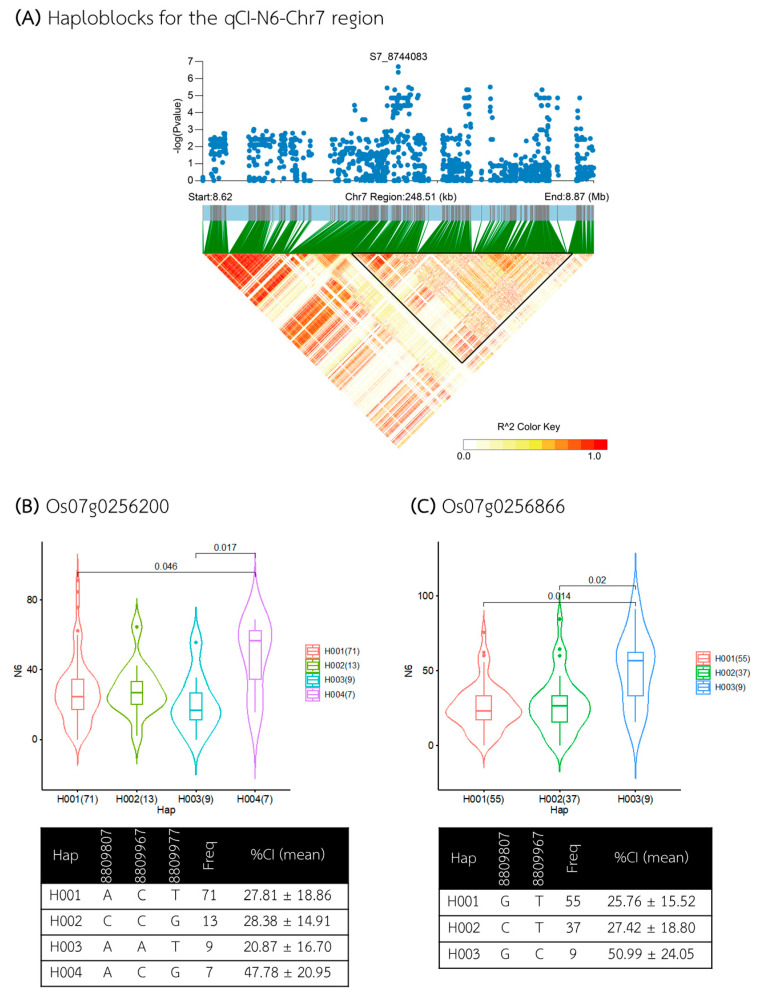
Analysis of associated regions and haplotype analysis within the qCI–N6–Chr7 region. (**A**) Manhattan plot and LD heatmap across the 248 kb region surrounding the significant SNP. The black triangle in the LD heatmap indicates the candidate haploblock. (**B**,**C**) Boxplots showing the distribution of %CI and haplotype analysis for the genes *Os07g0256200* and *Os07g0256866*.

**Figure 11 plants-13-02112-f011:**
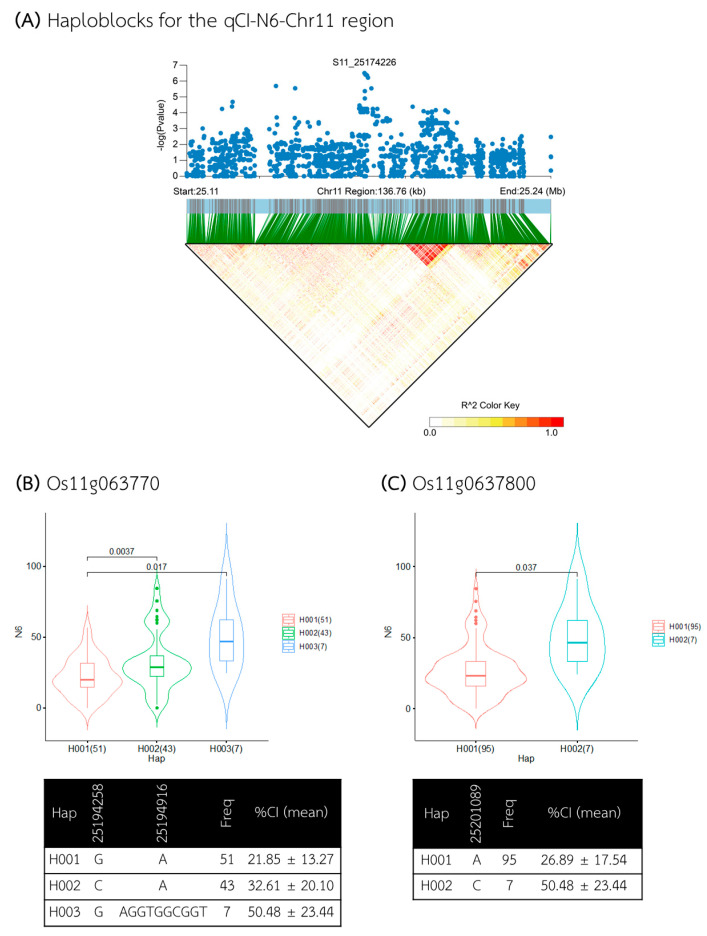
Analysis of associated regions and haplotype analysis of the qCI–N6–Chr11 region. (**A**) Manhattan plot and LD heatmap across the 136 kb region surrounding the significant SNP. The black triangle in the LD heatmap indicates the candidate haploblock. (**B**,**C**) Boxplots showing the distribution of %CI and haplotype analysis for the genes *Os11g0637700* and *Os11g0637800*.

**Table 1 plants-13-02112-t001:** The single-nucleotide polymorphisms (SNPs) that are associated with callus induction (CI) in three tissue culture media and annotated genes are located in the 400 kb regions.

Medium	QTL	SNP id.	Chr	Pos	*p*-Value	MAF	Effect
B5	qCI–B5–Chr6	S6_8152623	6	8,152,623	1.75 × 10^−6^	0.17	−13.81
MS	qCI–MS–Chr2	S2_8960153	2	8,960,153	4.74 × 10^−6^	0.13	−13.38
	qCI–MS–Chr6	S6_16813533	6	16,813,533	2.19 × 10^−6^	0.06	23.91
N6	qCI–N6–Chr6.1	S6_3503834	6	3,503,834	1.67 × 10^−6^	0.17	15.15
	qCI–N6–Chr6.2	S6_8003634	6	8,003,634	2.20 × 10^−6^	0.26	−10.82
	qCI–N6–Chr7	S7_8744083	7	8,744,083	2.00 × 10^−7^	0.21	16.03
	qCI–N6–Chr11	S11_25174226	11	25,174,226	3.13 × 10^−7^	0.47	9.94

**Table 2 plants-13-02112-t002:** Summary of the identified genes within each QTL.

QTLs	Block Start (bp)	Block End (bp)	Block Size (bp)	#SNPs	#Genes	#Candidate Genes	Annotation of Candidate Genes
qCI–B5–Chr6	7,934,342	8,198,412	264,070	56	21	16	*Os06g0252800*, *Os06g0253100*, *Os06g0253350*, *Os06g0253600*, *Os06g0254200*, *Os06g0254300*, *Os06g0254600*, *Os06g0255200*, *Os06g0255700*, *Os06g0255900*, *Os06g0256000*, *Os06g0256500*, *Os06g0256600*, *Os06g0256800*, *Os06g0257050*, *Os06g0257200*
qCI–MS–Chr2	8,906,794	8,968,876	62,082	12	4	0	–
qCI–MS–Chr6	16,575,366	16,981,445	406,079	31	13	0	–
qCI–N6–Chr6.1	3,483,450	3,555,107	71,657	7	4	3	*Os06g0169600*, *Os06g0169800*, *Os06g0170500*
qCI–N6–Chr6.2	7,995,999	8,014,002	18,003	4	1	1	*Os06g0254600*
qCI–N6–Chr7	8,734,157	8,858,375	124,218	11	5	2	*Os07g0256200*, *Os07g0256866*
qCI–N6–Chr11	25,104,139	25,244,312	140,173	56	13	10	*Os11g0636900*, *Os11g0637050*, *Os11g0637000*, *Os11g0637100*, *Os11g0637200*, *Os11g0637600*, *Os11g0637700*, *Os11g0637800*, *Os11g0637900*, *Os11g0638200*

## Data Availability

All relevant data have been provided as Tables and Figures in the text and Appendix A.

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
