# Peer review of "Genome-Wide Association Analysis Identifies Candidate Loci for Callus Induction in Rice (Oryza sativa L.)"

_plants, 2024, doi:10.3390/plants13152112_

Round 1

Reviewer 1 Report

Comments and Suggestions for Authors

Comments

Comments and Suggestions for Authors

Dear Author,

It is my pleasure to review the manuscript entitled “Genome-Wide Association Analysis Identifies Candidate Loci for the Callus Induction in Rice (Oryza sativa L.)” a research article submitted to MDPI Journal, Plants. Authors of this manuscript characterized callus induction (CI) features of 110 rice accessions using B5, MS and N6 culture media and analyzed the results via GWAS. They have identified several QTLs and responsive genes related to CI in rice.

The overall experiments, they performed, are well and the results are very convincing and important for rice cultivation. Thus, the presented results take up an important topic consistent with the profile of the Journal.

I have some suggestions, which might improve the manuscript to make important to the wider readers.

·         Improvement in English is necessary for clear understanding

·         Introduction should be more constructive and specific with rationale of the study. Elaborate clearly, why this research is necessary.

·         Include some more related recent work similar to your work even in other plants

·         Demonstrate their lacking and limitations and prospects of your research

·         Please check the original PDF where I made comments

·         Similarity percentage should be less than 20%

Results

Ø  You could supply original callus pictures of some of the accessions

Discussion is mostly repetition of results. Need rewritten

Materials

Ø  All rice may not have 100% germination rate. What was your selection criteria? From how many accessions you found 110 accessions with 100% germination rate? Or your research material all of them demonstrated 100% germination rate? You need to clear this process. Otherwise it is ambiguous that all materials showed 100% efficiency.

Ø  There is no genotype study

Ø  The main features of this study is phenotyping of the callus. However, you have just mentioned “callus induction frequency was recorded as a percentage”. No other process has been described. For research repeatability, you must describe the basic procedures. What was the criteria for callus induction, callus size, callus compactness, color, so many factors are related. Need description or references. You may follow your reference 19

Ø  You did not mention the germination percentage calculation process

Ø  No real experiment with any of the identified genes

Comments on the Quality of English Language

Author Response

Response to Reviewer 1

Comments and Suggestions for Authors

Dear Author,

It is my pleasure to review the manuscript entitled “Genome-Wide Association Analysis Identifies Candidate Loci for the Callus Induction in Rice (Oryza sativa L.)” a research article submitted to MDPI Journal, Plants. Authors of this manuscript characterized callus induction (CI) features of 110 rice accessions using B5, MS and N6 culture media and analyzed the results via GWAS. They have identified several QTLs and responsive genes related to CI in rice.

The overall experiments, they performed, are well and the results are very convincing and important for rice cultivation. Thus, the presented results take up an important topic consistent with the profile of the Journal.

I have some suggestions, which might improve the manuscript to make important to the wider readers.

  • Improvement in English is necessary for clear understanding

Answer: The English has been improved, as the certificate is attached to the Plants Editor.

  • Introduction should be more constructive and specific with rationale of the study. Elaborate clearly, why this research is necessary.

Answer: The introduction part has been rewritten in the revised manuscript.

  • Include some more related recent work similar to your work even in other plants

Answer: Related works about callus induction in grape and sorghum have been included in the revised manuscript (line number 58 – 63). In addition, works about GWAS are also included in the revised manuscript (line number 100 – 114)

  • Demonstrate their lacking and limitations and prospects of your research

Answer: The lack and limitations of previous research and the prospects of this research were incorporated in the introduction section of the revised manuscript.

  • Please check the original PDF where I made comments

Answer: The comments in the PDF file have been answered separately in another response file.

  • Similarity percentage should be less than 20%

Answer: The similarity of the revised manuscript was checked, and the similarity is 15%. Most of them were found in the Reference section.

Results

Ø  You could supply original callus pictures of some of the accessions

Answer: The images in callus induction have been illustrated in Figure 1 in the revised manuscript.

Discussion is mostly repetition of results. Need rewritten

Answer: The discussion section has been rewritten so as not to be repetitive with the Results section in the revised manuscript. A paragraph describing the common QTL for B5 and N6 on rice chromosome 6 was included in the revised manuscript's Discussion section (Lines 451 – 459).

Materials

Ø  All rice may not have 100% germination rate. What was your selection criteria? From how many accessions you found 110 accessions with 100% germination rate? Or your research material all of them demonstrated 100% germination rate? You need to clear this process. Otherwise it is ambiguous that all materials showed 100% efficiency.

Answer:

  1. The paragraph and supplementary table (Table S1) describing the rice accession germination experiment were included in the Results section (Line number 128 – 139).
  2. The paragraph describing the methodology of the germination experiment was included in the Materials and Methods section (Line number 462 - 467)

Ø  There is no genotype study

Answer: The genotype study of the candidate genes has not yet been included in this manuscript. We plan to narrow down the candidate genes further. Once the most likely candidate genes were identified, we will further develop functional markers for the callus induction trait and test their effiencies in the segregating populations. We have stated this statement in the revised manuscript's Conclusion section (Lines 515 – 518).

Ø  The main features of this study is phenotyping of the callus. However, you have just mentioned “callus induction frequency was recorded as a percentage”. No other process has been described. For research repeatability, you must describe the basic procedures. What was the criteria for callus induction, callus size, callus compactness, color, so many factors are related. Need description or references. You may follow your reference 19

Answer: The description of the callus induction experiment has been incorporated in the revised manuscript (Lines 467 – 477) so that the reader can repeat the experiment accordingly.

Ø  You did not mention the germination percentage calculation process

Answer: The germination experiment was described in the Materials and Methods section, and Supplementary Table S1 was added in the revised manuscript.

Ø  No real experiment with any of the identified genes

Answer: We plan to narrow down the candidate genes further. Once the most likely candidate genes were identified, we will further analyze the candidate genes' expression.  We have stated this statement in the revised Conclusion section (Lines 515 – 518) of the revised manuscript. 

Reviewer 2 Report

Comments and Suggestions for Authors

Introduction – row41-53 - provides only very general and superficial information about callus and the background of callogenesis. This part needs to be revised/rewritten with a focus on rice callus cultures.

Row 59 - you probably mean chromosome 3 in rice.

Objectives of the study should be better described, focusing not on the culture media but on the key components of these media.

Results:

chapter 2.1 – comparison of regeneration ability on 3 different culture media - Figure 1: Very poor quality and unreadable images. Histograms must be presented with a high-quality image and boxplots are not informative. It would be appropriate to replace them with one output from a statistical software - e.g., two-way ANOVA (genotype and media type).

Figure 2 - The text in the image is completely unreadable! Image 2B is redundant (and unreadable), the same information is on Fig. 2C.

How to explain that QTLs do not contain any candidate genes? Please explain.

Figure 5-10 - The graphic quality is good, but the text is unreadable. It would be appropriate to revise the graphical outputs and choose a uniform format for displaying the results.

Discussion: the information in the first paragraph is general and does not relate to the subject of the publication. Row301-303 - citation is missing. The influence of N6 medium is mentioned in the second paragraph, B5 medium in the fifth, then MS medium, and again N6 medium. The discussion must be rewritten and properly and logically organized.

The description of candidate genes and proteins involved in callus induction mechanisms is described in detail and well.

Conclusions - it would be appropriate to reformulate it and omit the citations

Author Response

Response to Reviewer 2

Comments and Suggestions for Authors

Introduction – row41-53 - provides only very general and superficial information about callus and the background of callogenesis. This part needs to be revised/rewritten with a focus on rice callus cultures.

Answer: A paragraph describing callus induction in rice has been included in the revised manuscript, Line number 69 – 89.

Row 59 - you probably mean chromosome 3 in rice.

Answer: The sentence was re-written as “Wu et al. (2022) showed that Pseudo-response Regulators (PRR) located on rice chromosome 3 regulated rice callus formation [19]; this gene reportedly involved circadian clock components [20].” In the revised manuscript, Line number 86 – 88.

Objectives of the study should be better described, focusing not on the culture media but on the key components of these media.

Answer: The description of the critical components of media used in this study has been added in the Introduction section (Lines 69 – 79), and the study's objective was also re-written, as found in Lines 122 – 126 of the revised manuscript.

Results:

chapter 2.1 – comparison of regeneration ability on 3 different culture media - Figure 1: Very poor quality and unreadable images. Histograms must be presented with a high-quality image and boxplots are not informative. It would be appropriate to replace them with one output from a statistical software - e.g., two-way ANOVA (genotype and media type).

Answer: Figure 2 of the revised manuscript has been changed to the new one with higher quality, both the histograms and informative boxplot. We also include the correlation plots of the three traits in Figure 2 of the revised manuscript.

Figure 2 - The text in the image is completely unreadable! Image 2B is redundant (and unreadable), the same information is on Fig. 2C.

Answer: Figure 3 of the revised manuscript has been changed to be readable and informative.

How to explain that QTLs do not contain any candidate genes? Please explain.

Answer: We added more results about candidate gene identification of the qCI-MS-Chr2 and qCI-MS-Chr6 in the Results section (line number 262 – 275) and also made more discussion about this issue in the Discussion section (Line number 404 – 414) in the revised manuscript.

Figure 5-10 - The graphic quality is good, but the text is unreadable. It would be appropriate to revise the graphical outputs and choose a uniform format for displaying the results.

Answer: Figures 6 – 11 of the revised manuscript have been improved to be readable and have a uniform format.

Discussion: the information in the first paragraph is general and does not relate to the subject of the publication. Row301-303 - citation is missing. The influence of N6 medium is mentioned in the second paragraph, B5 medium in the fifth, then MS medium, and again N6 medium. The discussion must be rewritten and properly and logically organized.

Answer:

  1. The first paragraph of the discussion section was rewritten to be more related to the present work (Lines 356 – 369).
  2. The reference of this work has been added to the Reference section of the revised manuscript as “37. Mekprasart, W., & Chutipaijit, S. (2023). Enhanced Efficiency in Plant Regeneration of Thai Rice Variety (Pathumthani1) Using Nano-carbon Materials Application. Journal of Advanced Development in Engineering and Science, 9(24), 16–26.”
  3. The Discussion section has been reorganized as suggested by the reviewer. A paragraph describing the common QTL for B5 and N6 on rice chromosome 6 was included in the Discussion section (Lines 451 – 459) in the revised manuscript.

The description of candidate genes and proteins involved in callus induction mechanisms is described in detail and well.

Answer: Thank you for your kind review of our work.

Conclusions - it would be appropriate to reformulate it and omit the citations

Answer: The Conclusion section was reformulated (Lines 510 – 524), and the citations were omitted, as suggested by the reviewer in the revised manuscript.

Reviewer 3 Report

Comments and Suggestions for Authors

This manuscript carried out a genome-wide association study (GWAS) analysis on 110 indica rice accessions with various callus induction trait, and it will provide desirable genes for rice transformation. The followings are suggested for authors in revising the manuscript.

 (i) The manuscript indicated 2,385,475 SNP markers were used for GWAS analysis, and the authors should provide the source of SNP data.

(ii) The authors should pay more attentions to the meaning of genes and proteins. Some genes were annotated as proteins, for examples, ‘several genes, including the beta-tubulin, the zinc finger proteins, the RNP-1 domain-containing protein’ (line 29-30), ‘Os06g0255200 is annotated as a conserved hypothetical protein’ (line 181), ‘These genes were annotated similarly to beta-tubulin (fragment) (Os06g0169600), the hypothetical protein’ (line 218-219), ‘Os06g0255200 and Os06g0255700 were chromatin remodeling proteins’ (line 340-341), ‘Os06g0169600, annotated as the beta-tubulin containing GTPase domain’ (line 366), ‘(B) Os06g0254600 (Caleosin related family protein’ (Figure8), etc.

(iii) The scientific names of plants should be italic letters, such as ‘Oryza sativa’ (line 3), ‘japonica and indica’ (line 39), ‘japonica and indica’ (line 56), ‘O. sativa’ (line 135), etc.

(iv) The ‘103 accessions’ in Figure 2 should be ‘110 accessions’ as showed in this figure legend.

(v) The ‘Kb’ in the text should be ‘kb’.

(vi) Table 2 should be a three-line table.

Author Response

Response to reviewer 3

This manuscript carried out a genome-wide association study (GWAS) analysis on 110 indica rice accessions with various callus induction trait, and it will provide desirable genes for rice transformation. The followings are suggested for authors in revising the manuscript.

(i) The manuscript indicated 2,385,475 SNP markers were used for GWAS analysis, and the authors should provide the source of SNP data.

Answer: The source of SNP data used in this study was provided in the Materials and Methods section, lines 481 – 483 of the revised manuscript.

(ii) The authors should pay more attentions to the meaning of genes and proteins. Some genes were annotated as proteins, for examples, ‘several genes, including the beta-tubulin, the zinc finger proteins, the RNP-1 domain-containing protein’ (line 29-30), ‘Os06g0255200 is annotated as a conserved hypothetical protein’ (line 181), ‘These genes were annotated similarly to beta-tubulin (fragment) (Os06g0169600), the hypothetical protein’ (line 218-219), ‘Os06g0255200 and Os06g0255700 were chromatin remodeling proteins’ (line 340-341), ‘Os06g0169600, annotated as the beta-tubulin containing GTPase domain’ (line 366), ‘(B) Os06g0254600 (Caleosin related family protein’ (Figure8), etc.

Answer: We have clarified all the genes mentioned in the revised manuscript to be not confuse with protein as follow;

- Moreover, several genes, including those that encode the beta-tubulin protein, the zinc finger protein, the RNP-1 domain-containing protein, and Lysophosphatidic acid acyltransferase, were associated with different CI percentages in the N6 medium. (Lines 29 – 31)

- “Notably, the gene Os06g0254600, which encodes a Caleosin-related family protein, was identified in the B5 and N6 media,” (Lines 222 – 223)

- “Os06g0255200 is annotated as a gene encoding conserved hypothetical protein,” (Line 244)

- “Os06g0254200 was annotated as a gene similar to potassium channel protein NKT5.” (Lines 246 – 247)

“two genes annotated to encode Caleosin-related family proteins,” (Line 247)

- “These genes were annotated as encoding the protein similar to beta-tubulin (Os06g0169600), the hypothetical protein (Os06g0169800), and similar to RNA-binding protein (Os06g0170500), respectively.” (Line 289 – 291)

- “Os06g0254600, which encodes a Caleosin-related family protein,” (Line 302)

- “the genes with the domain of unknown function DUF231), Os07g0256300 and Os07g0256866 (the genes encoding hypothetical protein), and Os07g0256200 (the gene with the RNA recognition motif, RNP-1 domain-containing protein).” (Lines 315 -318)

- “Os11g0638200 (annotated as the genes encoding a hypothetical protein); Os11g0637000, Os11g0637100, and Os11g0637200 (encoding monosaccharide transporter (PLT subfamily), PLT protein 10); Os11g0637600 (encoding protein similar to potyvirus VPg interacting protein); Os11g0637700 (encoding RNA-binding protein); and Os11g0637800 (Lysophosphatidic acid acyltransferase 2).” (Line 338 -342)

- “Os06g0254300 and Os06g0254600, were related to Caleosin; Os06g0255200 and Os06g0255700 were the gene-encoding chromatin remodeling proteins.” (Lines 396 – 398)

-“ the gene encoding beta-tubulin containing GTPase domain,” (Line 420)

- “the OsC3H40 gene, encodes zinc finger proteins” (Line 425)

- etc.

(iii) The scientific names of plants should be italic letters, such as ‘Oryza sativa’ (line 3), ‘japonica and indica’ (line 39), ‘japonica and indica’ (line 56), ‘O. sativa’ (line 135), etc.

Answer: All the scientific names were italicized in the revised manuscript.

(iv) The ‘103 accessions’ in Figure 2 should be ‘110 accessions’ as showed in this figure legend.

Answer: The number “103” was deleted from Figure 3B of the revised manuscript.

(v) The ‘Kb’ in the text should be ‘kb’.

Answer: All ‘Kb’ in the text was changed to ‘kb’ in the revised manuscript.

(vi) Table 2 should be a three-line table.

Answer: Table 2 was re-formatted as the three-line table in the revised manuscript.

Round 2

Reviewer 1 Report

Comments and Suggestions for Authors

The article has been improved substantially. 

Comments on the Quality of English Language

English is fine

Reviewer 2 Report

Comments and Suggestions for Authors

the manuscript has been fundamentally revised, missing data have been added, and the results of calogenesis are well documented

the discussion and conclusion were also rewritten and new information was added